# ZipAct: Zipping Interaction History into a Compact State for Efficient LLM Agents

**Zhiming Pan** *thomas_p@stu.pku.edu.cn*
*Peking University*

**Junyu Luo** *luojunyu@stu.pku.edu.cn*
*Peking University*

**Zhiping Xiao**\* *patricia.xiao@gmail.com*
*University of Washington*

**Kaize Ding** *kaize.ding@northwestern.edu*
*Northwestern University*

**Xiao Luo**\* *xiao.luo@wisc.edu*
*University of Wisconsin–Madison*

**Ming Zhang**\* *mzhang_cs@pku.edu.cn*
*Peking University*

**Reviewed on OpenReview:** `https://openreview.net/forum?id=ZssIalqqrz`

## Abstract

Current large language model (LLM) agentic frameworks typically rely on the entire raw interaction history to make decisions. Despite recent remarkable progress, this paradigm notably suffers from the *context snowball* effect: as the task progresses step by step, the history grows unboundedly, resulting in excessive token consumption and diluted agent attention. Toward this end, this paper proposes a novel and lightweight framework named `ZipAct`, which "zips" the lengthy history into a compact state during agentic reasoning. In particular, instead of feeding the full history to the model, our `ZipAct` maintains a structured state table comprising the agent's goal, world status and key constraints, which are updated dynamically at each step. Our simple design shifts agentic reasoning from a history-dependent paradigm to a state-dependent paradigm, which significantly reduces computational cost from quadratic ($O(T^2)$) to linear ($O(T)$) under a bounded-state assumption. Extensive, comprehensive experiments across multiple benchmark datasets demonstrate that `ZipAct` drastically reduces token usage while stably preserving or improving success rates compared to competing baselines. For reproducing results, our codebase can be accessed at: `https://github.com/Thomas-mci-21/ZipAct_TMLR`.

## 1 Introduction

Large language model (LLM) agents have demonstrated remarkable potential in solving complex, long-horizon tasks (Xi et al., 2023), ranging from embodied activities (Shridhar et al., 2021) to scientific reasoning (Wang et al., 2022) and web navigation (Yao et al., 2022). However, these tasks require extensive multi-turn interactions, leading to significant context growth and diluted agent attention. Consequently, a fundamental challenge arises: how to enable agents to handle unboundedly growing interaction histories within finite context windows without compromising efficiency or accuracy.

---

\*Corresponding authors.

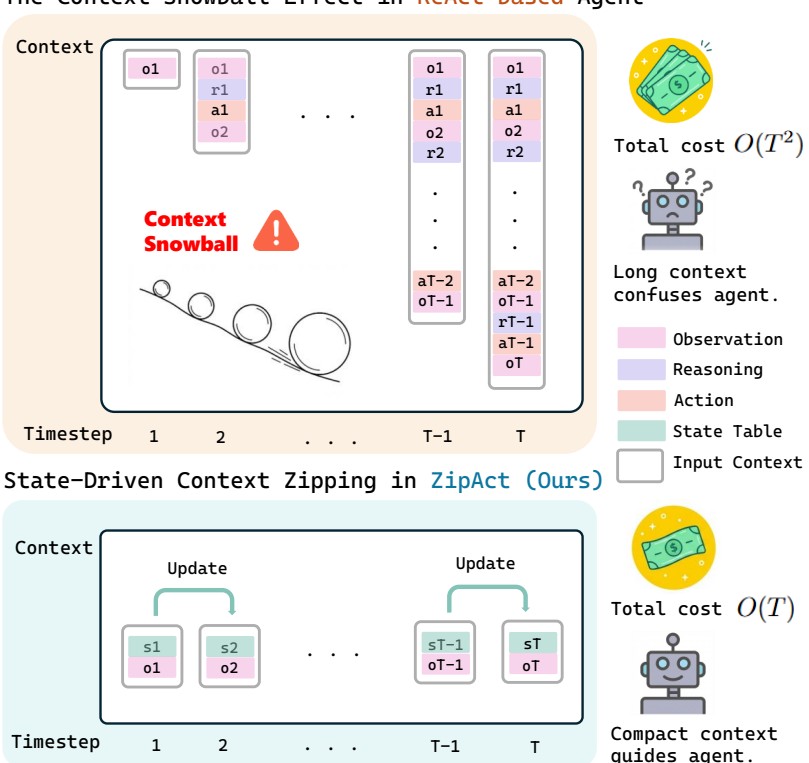

Figure 1: **ReAct vs. ZipAct.** ReAct accumulates raw history, leading to the context snowball effect. `ZipAct` zips history into a compact state, effectively eliminating redundancy and distraction.

Prevalent frameworks such as ReAct (Yao et al., 2023b) use the entire interaction history as context, concatenating all past observations and actions as input at each step. Even recent advancements such as ReflAct (Kim et al., 2025) and StateAct (Rozanov & Rei, 2025), which introduce self-reflection or state descriptions, typically append these features to the ever-growing history. While effective for short trajectories, these "append-only" approaches suffer from a critical phenomenon we term *context snowball*, creating a dual bottleneck (see Figure 1). First, the cumulative token consumption increases quadratically with the number of turns $T$, making long-horizon tasks prohibitively expensive (Liu et al., 2024b). Second, the accumulation of raw history introduces substantial noise; irrelevant details from early steps often obscure critical information, inducing hallucination or losing track of the immediate sub-goal (Liu et al., 2024a; Weston & Sukhbaatar, 2023). Although attempts like token masking (Lindenbauer et al., 2025) or compression via distillation (Kang et al., 2025) have been proposed, they either risk losing critical information or require expensive training. This necessitates a fundamental shift from passively recording history to actively managing state.

To bridge this gap, we propose **ZipAct** (see Figure 1), a lightweight, training-free framework that shifts agent reasoning from a history-dependent to a state-dependent paradigm. **ZipAct** maintains a compact structured state $S_t$ that "zips" the verbose history into three components: the *Goal State* ($\mathcal{G}$) for tracking hierarchical task progress, the *World State* ($\mathcal{W}$) for abstracting key environmental observations, and the *Constraint State* ($\mathcal{C}$) for managing action validity and feedback. Structurally, we decouple the system into a memory-less Actor, which decides actions based solely on the current state $S_t$, and a State Updater, which synthesizes the semantic transition from $t$ to $t+1$. Under a bounded-state assumption, this architecture keeps the Actor input compact and decoupled from the interaction history, reducing the cumulative inference cost from quadratic ($O(T^2)$) to linear ($O(T)$).

We evaluate **ZipAct** using both open-source (Qwen-2.5-7B/32B-Instruct (Yang et al., 2024a)) and proprietary (GPT-4o-mini/4o (OpenAI, 2024)) LLMs. We conduct extensive experiments across three representative benchmarks: ALFWorld (Shridhar et al., 2021), ScienceWorld (Wang et al., 2022), and WebShop (Yao

et al., 2022). Empirical results show that `ZipAct` substantially mitigates the context snowball effect over the evaluated benchmarks. Specifically, `ZipAct` reduces token consumption by 60.8%–67.6% across datasets while preserving or even improving the success rate compared to full-history baselines. With linear complexity ($O(T)$) under a bounded-state assumption, `ZipAct` provides an efficient alternative to accumulating raw history.

Our contribution is summarized as follows:

❶ *New Perspective.* We identify *context snowballing* as a fundamental bottleneck in existing LLM agent paradigms, stemming from unbounded history accumulation that incurs quadratic computational overhead and dilutes agent attention. We present `ZipAct`, a training-free framework that shifts agentic reasoning from passively recording trajectories to actively managing a compact, structured state.

❷ *Novel Methodology.* We propose a unified Goal-World-Constraint (G-W-C) schema within an Actor–Updater architecture, where the structured state $S_t = \langle \mathcal{G}_t, \mathcal{W}_t, \mathcal{C}_t \rangle$ tracks hierarchical task progress, denoised environment observations, and anti-loop constraints. This design reduces cumulative token complexity from quadratic ($O(T^2)$) to linear ($O(T)$) under a bounded-state assumption by decoupling inference cost from trajectory length.

❸ *Empirical Validation.* Extensive experiments across three benchmarks (ALFWorld, ScienceWorld, WebShop) and four backbone LLMs demonstrate that `ZipAct` is: (1) *highly efficient*, reducing token consumption by 60.8%–67.6%; (2) *performance-preserving or improving*, outperforming all cost-reduction baselines and remaining competitive with Reflexion at a fraction of its overhead; and (3) *model-agnostic*, achieving favorable trade-offs across all evaluated settings.

## 2 Problem Analysis

### 2.1 Preliminaries

We formulate the agent-environment interaction as a Partially Observable Markov Decision Process (POMDP), defined as $\mathcal{M} = \langle \mathcal{S}, \mathcal{A}, \mathcal{O}, \mathcal{P}, \mathcal{R}, \mathcal{G} \rangle$, where $\mathcal{S}$ denotes the latent state space (e.g., the full status of a household), which is not fully visible to the agent. $\mathcal{A}$ is the discrete action space (e.g., `open drawer 1`). $\mathcal{O}$ is the observation space, where $o_t \in \mathcal{O}$ provides a partial view of the current state $s_t$ at step $t$. $\mathcal{P}$ represents the transition dynamics, and $\mathcal{R}$ is the reward function. $\mathcal{G}$ is the natural language goal (e.g., "put a clean lettuce in dining table"). The agent's objective is to execute a sequence of actions that successfully completes the assigned task $\mathcal{G}$ within a limited number of steps.

**The ReAct Paradigm.** The dominant framework for LLM agents, exemplified by ReAct (Yao et al., 2023b) and its numerous variants (e.g., ReflAct (Kim et al., 2025)), relies on *interleaved reasoning and acting*. To facilitate planning, these methods introduce a *thought space* $\mathcal{T}$, where a thought $\tau_t \in \mathcal{T}$ is a natural language reasoning trace generated before the action. At each time step $t$, the agent's decision is conditioned on the entire history $H_t$, which accumulates all past observations, thoughts, and actions:

$$H_t = [\mathcal{G}, o_0, \tau_0, a_0, o_1, \tau_1, a_1, \ldots, o_t]. \tag{1}$$

The policy $\pi$ operates in a cycle: first generating a thought $\tau_t \sim \pi(\cdot|H_t)$, appending it to the context, and then predicting an action $a_t \sim \pi(\cdot|H_t, \tau_t)$.

While this existing paradigm effectively grounds the model's reasoning, Eq. 1 reveals a critical structural flaw: the input context $H_t$ grows linearly with $t$. As we specifically analyze in the next section, this context snowballing introduces quadratic computational complexity and accumulates distracting historical noise, fundamentally limiting the efficiency and robustness of long-horizon agents.

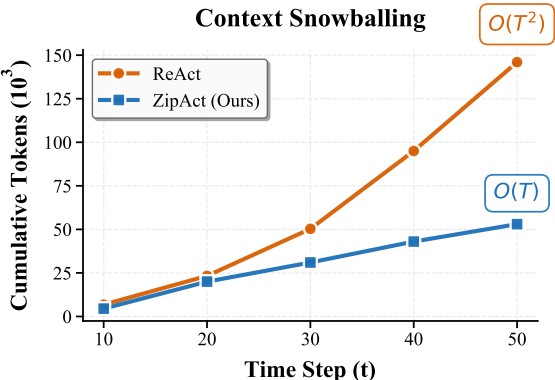

Figure 2: **The Context Snowball Effect.** ReAct (orange) shows quadratic growth ($O(T^2)$), while `ZipAct` (blue) maintains linear complexity ($O(T)$).

## 2.2 The Context Snowball Challenge

In this section, we analyze the limitations of history-dependent architectures (e.g., ReAct (Yao et al., 2023b)) from two perspectives: computational efficiency and cognitive distraction. We term the phenomenon where the input context grows uncontrollably as *context snowballing*.

**Quadratic Complexity.** The standard practice of concatenating all past observations, thoughts, and actions imposes prohibitive costs in long-horizon tasks. Let $L_{\text{step}}$ denote the average number of tokens generated in one interaction cycle. For a task with horizon $T$, the length of the input context at step $t$ is $|C_t| \approx t \cdot L_{\text{step}}$. As a result, the cumulative token consumption $\mathcal{C}_{total}$ scales quadratically with the task horizon:

$$\mathcal{C}_{total} \approx \sum_{t=1}^{T} (t \cdot L_{\text{step}}) = O(T^2). \tag{2}$$

This quadratic growth is unsustainable. As illustrated in Figure 2 (orange line), the token usage of a ReAct agent explodes as the task progresses. For a 50-step task in ALFWorld, ReAct processes over 146k tokens in total, whereas a linear method would only require 53k. This results in high latency and deployment costs.

**Cognitive Load: Information Dilution.** Beyond computational costs, the "snowballing" history imposes a severe cognitive burden. In realistic environments, agents frequently execute long trajectories involving exploration and trial-and-error. However, ReAct-based frameworks indiscriminately retain every interaction detail, regardless of relevance. Consequently, critical signals required for correct reasoning (e.g., the original goal and current inventory) become increasingly obscured by an accumulation of redundant information. This aligns with the "Lost-in-the-Middle" phenomenon (Liu et al., 2024a), where LLMs struggle to retrieve key details from extended contexts. We argue that this unstructured accumulation dilutes the agent's focus. As illustrated in the ReAct failure case in Figure 3, the agent encounters a distractor ('soapbottle 2') while searching for a soapbar. The verbose logs from handling this distractor drown out the earlier observation that the cabinet does not contain the target item. Consequently, the agent forgets this negative constraint and enters a "hallucination loop," repeating the invalid search. As illustrated in the ReAct baseline example in Figure 3, as history grows, the model's attention to the initial goal diminishes, often precipitating "hallucination loops" where the agent repeats invalid actions.

## 3 Proposed Method: `ZipAct`

To mitigate the context snowballing effect inherent in history-dependent agents (e.g., ReAct), we introduce `ZipAct`. The core insight is to shift the agent's reasoning from a *history-dependent* paradigm to a *state-dependent* paradigm. Formally, rather than conditioning the policy on the ever-growing interaction trajectory

```
ALFWorld: put a clean soapbar in countertop.
```

```
...
Observation: You open the cabinet 1. The cabinet 1 is open. In it, you see
a soapbottle 2.
Thought: I found a soapbottle 2 in the cabinet 1, but I need a soapbar. I
should check if there's a soapbar in the cabinet 1 or other receptacles.
Since I already opened the cabinet 1, I can check its content again.
Action: take soapbottle 2 from cabinet 1

Observation: You pick up the soapbottle 2 from the cabinet 1.
Thought: I took a soapbottle 2, but I need a soapbar. I should try to find
a soapbar in other receptacles. Since I already have an object in my hand,
I can try to put it somewhere and then search for a soapbar.
Action: put soapbottle 2 in cabinet 1

Observation: You put the soapbottle 2 in/on the cabinet 1
Thought: I put the soapbottle 2 back in the cabinet 1. Now I can search for
a soapbar. I'll go to the cabinet 1 again to check if there's a soapbar.
Action: go to cabinet 1
                         ❌ Stuck in Loop
Observation: Nothing happens
Thought: Since "Nothing happened", my previous action was invalid. I should
try to go to cabinet 1 again, but this time, I should also check if there's
a soapbar in the cabinet 1.
Action: go to cabinet 1
                                            ReAct-based Agent
Observation: Nothing happens
```

```
→ The interactions with the distractor object (soapbottle 2)
diluted the context. The agent "forgot" the observation from
previous step and redundantly searches the same empty location.
```

Figure 3: **Information Dilution** in ALFWorld. Interactions with a distractor (soapbottle) dilute the context, causing the agent to forget the cabinet is empty and redundantly search the same location.

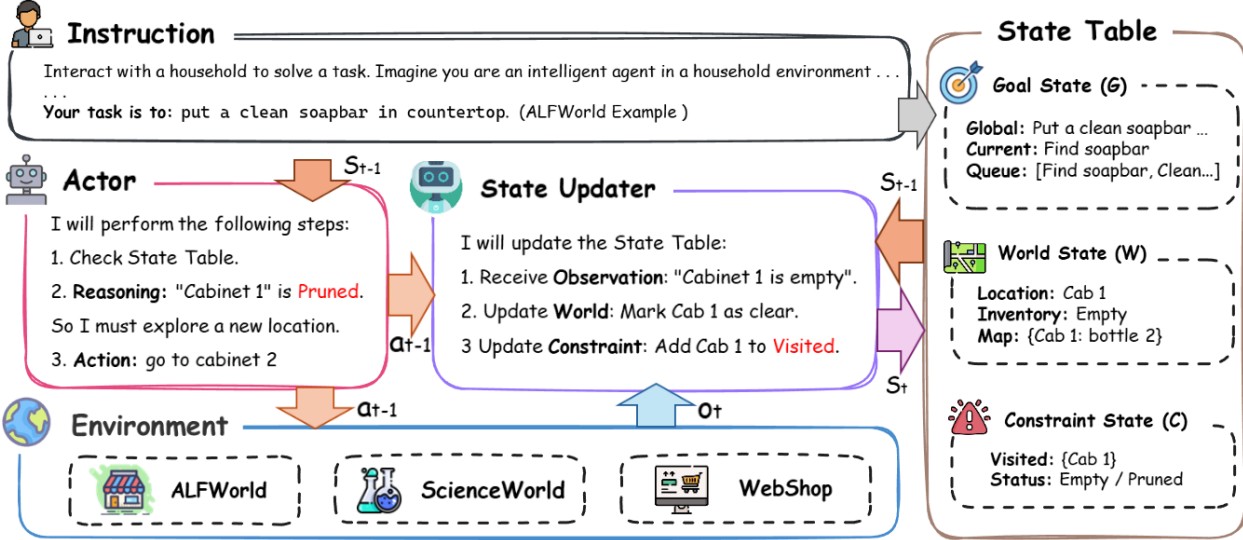

Figure 4: The overall framework of the proposed `ZipAct`. The State Updater synthesizes the interaction history into a compact State Table comprising Goal (G), World (W), and Constraint (C) states. The Actor generates actions based on this structured state, effectively decoupling inference cost from trajectory length.

$H_t$, `ZipAct` maintains a compact, structured state $S_t$ that explicitly tracks the task progress and environment status. The decision process is decoupled into two phases: state updating and action generation.

$$S_t = \mathcal{U}(S_{t-1}, a_{t-1}, o_t), \tag{3}$$

$$\tau_t, a_t \sim \pi(\cdot | S_t, o_t), \tag{4}$$

where $\mathcal{U}$ is the State Updater and $\pi$ is the Actor policy. Since the dimension of $S_t$ is designed to be fixed (or bounded) regardless of step $t$, the input length for the Actor remains constant ($O(1)$ context). Consequently, this architectural shift reduces the cumulative computational complexity from quadratic $O(T^2)$ to linear $O(T)$ under a bounded-state assumption, making long-horizon reasoning more efficient.

## 3.1 Framework Overview

Drawing inspiration from cognitive architectures for agents (Sumers et al., 2024; Xi et al., 2023), `ZipAct` adopts a modular design that decouples information synthesis from decision making. This architecture comprises two collaborative units (shown in Figure 4). The *Actor ($\pi$)* operates as a memory-less decision engine. Unlike standard agents that re-read the full history log, the Actor predicts the next action $a_t$ conditioned *solely* on the current synthesized state $S_t$ and the immediate observation $o_t$. The *State Updater ($\mathcal{U}$)*, acting as a semantic compressor, is responsible for the transition dynamics. It digests the raw feedback from the environment and "zips" the interaction history into a refreshed state $S_t$, explicitly filtering out noise while preserving actionable context.

## 3.2 Structured State Representation

A key challenge in discarding history is preserving the critical context required for valid reasoning. To address this, we define the state as a structured tuple $S_t = \langle \mathcal{G}_t, \mathcal{W}_t, \mathcal{C}_t \rangle$, where each component targets a specific dimension of the POMDP.

**Goal State ($\mathcal{G}_t$).** This component maintains the hierarchical consistency of the task, formalized as:

$$\mathcal{G}_t = \langle g_{\text{global}}, g_{\text{curr}}, Q_{\text{plan}} \rangle. \tag{5}$$

It is composed of a static global instruction $g_{\text{global}}$, a dynamic sub-goal queue $Q_{\text{plan}}$, and the current objective $g_{\text{curr}}$. Leveraging task decomposition strategies proven effective for long-horizon reasoning (Zhou et al., 2023; Khot et al., 2023), `ZipAct` isolates $g_{\text{curr}}$ to ensure the Actor remains grounded in the immediate step, while $Q_{\text{plan}}$ prevents the agent from losing track of the long-term objective.

**World State ($\mathcal{W}_t$).** Formally, we define:

$$\mathcal{W}_t = \langle l_t, I_t, O_{\text{map}} \rangle, \tag{6}$$

serving as a denoised environment snapshot that abstracts raw observations into structured semantic variables, akin to hierarchical world modeling (Zhu et al., 2023). It tracks logical location $l_t$, inventory $I_t$, and a selective entity map $O_{\text{map}}$ (e.g., {"Fridge": "Open"}). By prioritizing causal abstraction (Wu et al., 2023), $\mathcal{W}_t$ discards the noise of raw history, retaining only state changes causally relevant to future actions.

**Constraint State ($\mathcal{C}_t$).** To act as an explicit anti-loop mechanism, we define:

$$\mathcal{C}_t = \langle C_{\text{neg}}, V_{\text{visited}} \rangle, \tag{7}$$

recording negative constraints (e.g., "Drawer 1 is locked") and visited states respectively. By incorporating self-correction principles (Welleck et al., 2023; Madaan et al., 2023), this design prunes invalid search spaces, preventing cyclic failures without re-analyzing full history logs.

## 3.3 State Update Mechanism

To synthesize information within a bounded context, we instantiate the Updater $\mathcal{U}$ as a semantic compiler, employing standardized operating procedures (Hong et al., 2024) (see prompts in §3.4). At each step, it performs a transformation:

$$S_t \leftarrow \mathcal{U}(S_{t-1}, a_{t-1}, o_t) \tag{8}$$

via three logical flows. First, *goal progression analysis* updates $Q_{plan}$ if $o_t$ entails $g_{curr}$ successful completion. Second, *world patching* captures critical semantic changes in $\mathcal{W}_{t-1}$ via selective context retention (Li et al., 2023). Finally, *failure diagnosis* parses execution errors into persistent rules in $\mathcal{C}_t$, leveraging interactive

---

**Algorithm 1** Inference Algorithm of `ZipAct`

---

    **Input:** Instruction $\mathcal{I}$, Environment $\mathcal{E}$
    **Output:** Task Success Status (Success / Failure)
  1: **Initialize:** $S_0 \leftarrow \langle \mathcal{I}, \emptyset, \emptyset \rangle$;    $o_0 \leftarrow \mathcal{E}.\text{reset}()$
  2: **for** $t = 1$ **to** $T_{max}$ **do**
  3:     *// Phase 1: Actor (Thought & Action)*
  4:     $\tau_t, a_t \leftarrow \pi(S_{t-1}, o_{t-1})$
  5:     *// Phase 2: Execution*
  6:     $o_t, \_\_, done \leftarrow \mathcal{E}.\text{step}(a_t)$
  7:     **if** *done* **then**
  8:         **return success**
  9:     **end if**
10:     *// Phase 3: Updater (State Compression)*
11:     $\Delta_g, \Delta_w, \Delta_c \leftarrow \mathcal{U}(S_{t-1}, a_t, o_t)$
12:     $S_t \leftarrow \text{Update}(S_{t-1}, \langle \Delta_g, \Delta_w, \Delta_c \rangle)$
13: **end for**
14: **return failure**

---

tool-based verification (Gou et al., 2024). Crucially, this decouples state size from trajectory length $T$. Unlike ReAct's unbounded history where irrelevant context severely distracts the model (Shi et al., 2023), `ZipAct` maintains a compact state representation, yielding linear cumulative inference cost $O(T)$ under a bounded-state assumption.

### 3.4 Prompt Design

We detail `ZipAct`'s core prompts in Figure 5. The **State Updater Prompt** instructs the model to synthesize $(S_{t-1}, a_{t-1}, o_t)$ into a refreshed structured state via three logical flows (goal progression, world patching, failure reflection), outputting a structured JSON $(G, W, C)$ (full schema in Appendix F). The **Actor Prompt** enforces memory-less decision-making, grounding every action solely in the current state table and latest observation. Both templates are adapted per environment via domain-specific injectables (Table 1).

Table 1: Domain-specific injectables adapting the `ZipAct` prompt to each environment.

| Dataset | Specific Injectables (Examples) |
| --- | --- |
| **ALFWorld** | **Env:** Household. **Entities:** `{cup: dirty, fridge: open}`. **Actions:** `take, put, open, clean, heat, cool` |
| **SciWorld** | **Env:** Laboratory. **Entities:** `{beaker: {temp:100C}}`. **Actions:** `pour, mix, ignite, measure, wait` |
| **WebShop** | **Env:** E-Commerce. **Entities:** `current_page, cart_items`. **Actions:** `search[q], click[btn], click[buy]` |

## 4 Experiments

### 4.1 Experimental Settings

**Benchmarks.** We benchmark `ZipAct` on three representative text-based environments.

**ALFWorld** (Shridhar et al., 2021) is a text-based simulation derived from the embodied ALFRED dataset, designed to evaluate interpreting natural language instructions and executing multi-step household tasks (e.g., picking, placing, cleaning, heating objects). By converting vision-and-language challenges into purely textual form, it rigorously assesses high-level planning and environment grounding. Task completion is measured as a binary success signal.

**ScienceWorld** (Wang et al., 2022) evaluates scientific reasoning and procedural task completion grounded in K–12 curricula. Agents must conduct virtual experiments spanning physics, chemistry, and biology, requiring

```
┌─────────────────────────────────────────┐   ┌─────────────────────────────────────────┐
│ State Updater Prompt                     │   │ Actor Prompt                             │
├─────────────────────────────────────────┤   ├─────────────────────────────────────────┤
│ Role & Objective                         │   │ Constraint & Context                     │
│ Synthesize (S_{t-1}, a_{t-1}, o_t) → S_t │   │ Memory-less mode: decisions based solely │
│ for an agent in <ENV>, focusing on       │   │ on the State Table (G, W, C) and the     │
│ semantic state changes.                  │   │ latest observation o_t.                  │
├ ─ ─ ─ ─ ─ ─ ─ ─ ─ ─ ─ ─ ─ ─ ─ ─ ─ ─ ─ ─ ┤   ├ ─ ─ ─ ─ ─ ─ ─ ─ ─ ─ ─ ─ ─ ─ ─ ─ ─ ─ ─ ─ ┤
│ State Schema                             │   │ Decision Protocol                        │
│ • G: Hierarchical task progress.         │   │ 1. Constraints: Avoid negative_constraints│
│ • W: Location, Inventory, Object states. │   │ 2. Alignment: Advance current_objective. │
│ • C: Failures & visited locations.       │   │ 3. Grounding: Verify target exists at    │
├ ─ ─ ─ ─ ─ ─ ─ ─ ─ ─ ─ ─ ─ ─ ─ ─ ─ ─ ─ ─ ┤   │    location.                             │
│ Update Protocol                          │   ├ ─ ─ ─ ─ ─ ─ ─ ─ ─ ─ ─ ─ ─ ─ ─ ─ ─ ─ ─ ─ ┤
│ 1. Goal: Advance queue on sub-goal       │   │ Action Space: <ENV_PRIMITIVES>           │
│    completion.                           │   │ (e.g., take, put, go to, search[q],      │
│ 2. World: Patch location/inventory/      │   │ click[btn])                              │
│    entity_map.                           │   ├ ─ ─ ─ ─ ─ ─ ─ ─ ─ ─ ─ ─ ─ ─ ─ ─ ─ ─ ─ ─ ┤
│ 3. Constraint: Failures to               │   │ Output:                                  │
│    negative_constraints.                 │   │ Thought: <reasoning over State Table>    │
├ ─ ─ ─ ─ ─ ─ ─ ─ ─ ─ ─ ─ ─ ─ ─ ─ ─ ─ ─ ─ ┤   │ Action: <action command>                 │
│ Output: Structured JSON (G, W, C)        │   └─────────────────────────────────────────┘
│ (full schema in Appendix F).             │
└─────────────────────────────────────────┘
```

Figure 5: Core prompt templates for the **State Updater** (left) and **Actor** (right). The Updater compresses interaction history into a structured state; the Actor acts solely on that state without accessing raw history.

multi-step interactions, hypothesis testing, and causal inference in a dynamic laboratory setting. Performance is measured as binary task success rate.

**WebShop** (Yao et al., 2022) is a simulated e-commerce environment containing over 1.18 million products. It challenges agents to navigate search results and purchase items satisfying user-specified constraints (e.g., price range, product attributes), assessing proficiency in query formulation and sequential web interaction. Performance is measured as binary task success rate.

**Agent Models.** We evaluate **ZipAct** on four backbone LLMs: GPT-4o, GPT-4o-mini (OpenAI, 2024), Qwen-2.5-7B-Instruct, and Qwen-2.5-32B-Instruct (Yang et al., 2024a). These backbones allow us to test **ZipAct** across both proprietary and open-source model families, as well as across different model capacities. This setup evaluates whether **ZipAct** remains effective across diverse model choices.

**Evaluation Protocol.** All methods are run with a maximum of 50 interaction steps per episode, and episodes that do not finish within this budget are counted as failures. Unless otherwise noted, reported cumulative token counts include both prompt and completion tokens across all LLM calls. ReAct uses one action-generation call per interaction step, whereas ZipAct uses one initialization call, one Actor call per step, and one Updater call after each executed action from the second step onward. Additional evaluation and efficiency details are provided in Appendix B and Appendix C.

**Baselines.** We compare **ZipAct** with the following four baselines.

**ReAct** (Yao et al., 2023b) is the standard paradigm where the agent conditions every decision on the full raw interaction history $H_t$. It concatenates all past observations, thoughts, and actions without compression, serving as the primary baseline representing the context snowball effect.

**ReAct + Observation Masking** (Lindenbauer et al., 2025) modifies ReAct by selectively masking older observations while retaining the complete action history. By removing verbose environment descriptions from early turns, it reduces token consumption but risks severing semantic dependencies required for grounding. In our implementation, observations older than the most recent five are replaced with the literal placeholder [Observation masked].

Table 2: Comparison of **ZipAct** with ReAct, Mask Obs., Summary, and Reflexion across ALFWorld, ScienceWorld, and WebShop. SR denotes success rate. Token Reduction: average tokens saved vs. ReAct. **Bold** indicates best performance; underlined indicates second best. Arrows denote performance change relative to ReAct.

| Model | Method | Success Rate (SR) % | | | Average SR | Avg. Tokens (Reduction) |
|---|---|---|---|---|---|---|
| | | ALFWorld | ScienceWorld | WebShop | | |
| *GPT-4o-mini* | ReAct | 53.0 | 14.2 | 48.5 | 38.6 | 158k (Base) |
| | Mask Obs. | 47.0 $_{\downarrow 6.0}$ | 10.4 $_{\downarrow 3.8}$ | 47.2 $_{\downarrow 1.3}$ | 34.9 $_{\downarrow 3.7}$ | 88k (-44.3%) |
| | Summary | 50.7 $_{\downarrow 2.3}$ | 8.1 $_{\downarrow 6.1}$ | 40.9 $_{\downarrow 7.6}$ | 33.2 $_{\downarrow 5.4}$ | 82k (-48.1%) |
| | Reflexion | **66.4** $_{\uparrow 13.4}$ | 18.5 $_{\uparrow 4.3}$ | **55.2** $_{\uparrow 6.7}$ | **46.7** $_{\uparrow 8.1}$ | 221k (+39.9%) |
| | ZipAct | 61.9 $_{\uparrow 8.9}$ | **19.9** $_{\uparrow 5.7}$ | 50.3 $_{\uparrow 1.8}$ | 44.0 $_{\uparrow 5.4}$ | **62k (-60.8%)** |
| *GPT-4o* | ReAct | 85.1 | 30.5 | 62.4 | 59.3 | 163k (Base) |
| | Mask Obs. | 80.5 $_{\downarrow 4.6}$ | 32.2 $_{\uparrow 1.7}$ | 59.8 $_{\downarrow 2.6}$ | 57.5 $_{\downarrow 1.8}$ | 91k (-44.2%) |
| | Summary | 82.1 $_{\downarrow 3.0}$ | 29.4 $_{\downarrow 1.1}$ | 57.5 $_{\downarrow 4.9}$ | 56.3 $_{\downarrow 3.0}$ | 89k (-45.4%) |
| | Reflexion | **93.3** $_{\uparrow 8.2}$ | **42.0** $_{\uparrow 11.5}$ | **70.1** $_{\uparrow 7.7}$ | **68.5** $_{\uparrow 9.2}$ | 245k (+50.3%) |
| | ZipAct | 85.8 $_{\uparrow 0.7}$ | 32.7 $_{\uparrow 2.2}$ | 65.5 $_{\uparrow 3.1}$ | 61.3 $_{\uparrow 2.0}$ | **53k (-67.5%)** |
| *Qwen-2.5-7B* | ReAct | 61.5 | 8.5 | 42.1 | 37.4 | 154k (Base) |
| | Mask Obs. | 58.2 $_{\downarrow 3.3}$ | 4.3 $_{\downarrow 4.2}$ | 37.1 $_{\downarrow 5.0}$ | 33.2 $_{\downarrow 4.2}$ | 78k (-49.4%) |
| | Summary | 57.6 $_{\downarrow 3.9}$ | 3.8 $_{\downarrow 4.7}$ | 38.9 $_{\downarrow 3.2}$ | 33.4 $_{\downarrow 4.0}$ | 76k (-50.6%) |
| | Reflexion | 68.2 $_{\uparrow 6.7}$ | **10.2** $_{\uparrow 1.7}$ | **46.0** $_{\uparrow 3.9}$ | **41.5** $_{\uparrow 4.1}$ | 213k (+38.3%) |
| | ZipAct | **69.4** $_{\uparrow 7.9}$ | 5.2 $_{\downarrow 3.3}$ | 42.1 | 38.9 $_{\uparrow 1.5}$ | **58k (-62.3%)** |
| *Qwen-2.5-32B* | ReAct | 81.7 | 26.4 | 58.9 | 55.7 | 170k (Base) |
| | Mask Obs. | 82.8 $_{\uparrow 1.1}$ | 25.1 $_{\downarrow 1.3}$ | 51.3 $_{\downarrow 7.6}$ | 53.1 $_{\downarrow 2.6}$ | 80k (-52.9%) |
| | Summary | 78.3 $_{\downarrow 3.4}$ | 23.7 $_{\downarrow 2.7}$ | 54.7 $_{\downarrow 4.2}$ | 52.2 $_{\downarrow 3.5}$ | 82k (-51.8%) |
| | Reflexion | **89.5** $_{\uparrow 7.8}$ | **35.1** $_{\uparrow 8.7}$ | **64.2** $_{\uparrow 5.3}$ | **62.9** $_{\uparrow 7.2}$ | 233k (+37.1%) |
| | ZipAct | 86.6 $_{\uparrow 4.9}$ | 28.0 $_{\uparrow 1.6}$ | 59.7 $_{\uparrow 0.8}$ | 58.1 $_{\uparrow 2.4}$ | **55k (-67.6%)** |

**ReAct + Summarization** (Park et al., 2023; Zhong et al., 2023) periodically condenses the full interaction history into a concise natural language summary that replaces the raw log. This mitigates context growth but often loses fine-grained details critical for long-horizon planning. In our implementation, summarization is triggered every 10 interaction steps; full prompting and decoding details for both baselines are provided in Appendix D.

**Reflexion** (Shinn et al., 2023) incorporates a verbal reinforcement learning loop where the agent generates a self-reflective critique upon task failure, appending it to the context for subsequent attempts. This represents methods that augment context to improve reasoning quality at higher token costs.

## 4.2 Results

Table 2 compares **ZipAct** with ReAct, Mask Observation, Summary, and Reflexion across three benchmarks. In terms of efficiency, **ZipAct** achieves a substantial token reduction of 60.8%–67.6%, significantly outperforming heuristic approaches like Mask Obs. and Summary (∼45% reduction). Unlike these heuristics, which merely delay context saturation, **ZipAct** uses a compact state representation that substantially reduces the need to reprocess redundant history. In contrast, Reflexion incurs a heavy computational penalty, increasing token consumption by 37%–50% due to its verbose reasoning process. Regarding success rate, **ZipAct** outperforms the cost-reduction baselines across all backbone models (e.g., 61.3% vs. 57.5% average SR with GPT-4o), demonstrating that structured states preserve critical context—such as visited locations and inventory changes—better than simple pruning strategies. While Reflexion generally achieves the highest success rates via iterative self-correction, **ZipAct** remains a strong runner-up with a fraction of the computational cost.

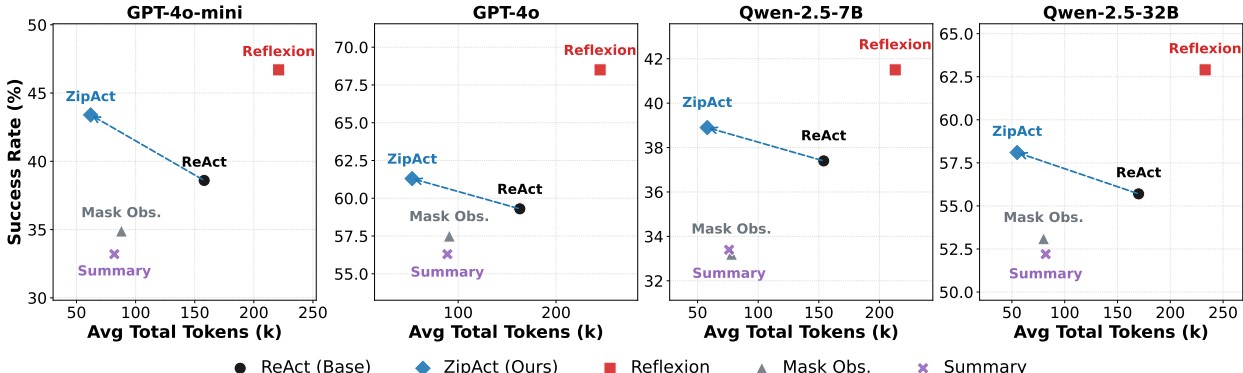

Figure 6: **Efficiency–Performance Trade-off** averaged across ALFWorld, ScienceWorld, and WebShop. **ZipAct** (ours) reduces token consumption by ∼60% while maintaining or improving success rates. Unlike heuristics (Mask/Summary) that degrade performance, **ZipAct** achieves a favorable balance, notably surpassing the full-history ReAct baseline on strong models (GPT-4o, Qwen-2.5-32B) by effectively filtering context noise.

Notably, with Qwen-2.5-7B on ALFWorld, **ZipAct** even surpasses Reflexion (69.4% vs. 68.2%). We attribute this to the limited attention span of smaller models; for them, eliminating noise via **ZipAct** reduces distraction and can be more effective than augmenting context with lengthy reflection.

These results suggest that the performance and efficiency gains arise from the combination of structured state organization and state-conditioned action generation. Under a bounded-state assumption, by decoupling inference costs from the trajectory length and empirically improving prompt-token scaling over the evaluated horizons, our framework substantially mitigates the context snowball effect observed in standard history-dependent agents. Furthermore, by filtering out irrelevant historical noise while explicitly tracking constraints, **ZipAct** mitigates the "Lost in the Middle" phenomenon, ensuring the agent remains grounded in the current subgoal throughout trajectories. Consequently, **ZipAct** achieves a favorable balance between efficiency and performance, providing a robust and scalable solution for environments with limited resources where the high latency of reflective methods would be prohibitive. At the same time, lower token usage does not necessarily translate directly into lower wall-clock latency, since end-to-end latency also depends on the number of model calls and provider-side overhead.

Table 3: Ablation studies of state components for **ZipAct** using Qwen-2.5-32B as the backbone. $G$: Goal State, $W$: World State, $C$: Constraint State. Removing $W$ leads to complete failure, highlighting its critical role in grounding.

| Method | $G$ | $W$ | $C$ | ALFWorld | SciWorld | WebShop | Avg. |
|---|---|---|---|---|---|---|---|
| V1 | ✓ | × | × | 0.0 | 0.0 | 0.0 | 0.0 |
| V2 | ✓ | ✓ | × | 53.6 | 10.3 | 22.9 | 28.9 |
| **ZipAct** | ✓ | ✓ | ✓ | **86.6** | **28.0** | **59.7** | **58.1** |

### 4.3  Further Analysis

**Ablation Study.** To validate the effectiveness of our structured state design, we compare **ZipAct** with two variants: (1) V1 (Goal Only), a minimal variant that maintains only the Goal State ($\mathcal{G}$) without tracking environment status ($\mathcal{W}$) or history constraints ($\mathcal{C}$); and (2) V2 (w/o Constraint), a variant that includes both Goal and World States ($\mathcal{G}, \mathcal{W}$) but omits the Constraint State ($\mathcal{C}$). Table 3 presents the results. First, removing the World State ($\mathcal{W}$) in V1 results in a complete failure (0.0% success rate) across all benchmarks. This indicates that goal tracking alone is insufficient: without a persistent grounded world state, the agent

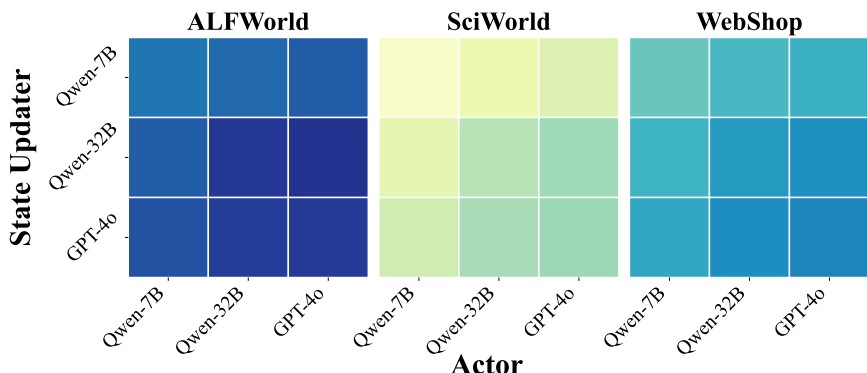

Figure 7: The Success Rate of **ZipAct** across ALFWorld, SciWorld, and WebShop when mixing different LLMs for the State Updater and Actor modules.

cannot reliably keep track of location, inventory, object status, or previously established world facts, which leads to invalid long-horizon action selection. Second, while incorporating $\mathcal{W}$ in V2 restores basic capability (28.9% average SR), the full **ZipAct** model further doubles the performance to 58.1%. This substantial gain confirms the critical role of the Constraint State ($\mathcal{C}$) as an anti-loop mechanism, which effectively prunes invalid actions and guides the agent through complex, long-horizon tasks. Overall, the complete G-W-C schema proves essential for robust reasoning.

**Efficiency-Performance Trade-off.** We visualize the token consumption versus success rate in Figure 6 to analyze the efficiency-performance spectrum. Existing paradigms typically fall into sub-optimal extremes: Reflexion achieves high success rates but incurs prohibitive computational costs (top-right), whereas heuristic pruning methods like Masking and Summary drastically reduce costs at the expense of performance (bottom-left). **ZipAct** breaks this dichotomy, occupying the ideal top-left region across all four evaluated backbones (GPT-4o series and Qwen-2.5 series). Compared to the standard ReAct baseline, **ZipAct** demonstrates a dominant improvement: it not only slashes token usage by 60.8% to 67.6% but also enhances success rates by 1.5% to 5.4%. This universal improvement validates that our structured state representation offers a superior mechanism for context management, decoupling inference cost from trajectory length without compromising reasoning accuracy. This substantial cost reduction makes deploying complex agents highly practical, even under strict API budgets or limited local memory.

**Impact of Model Specialization.** While Figure 6 uses a unified model for both components, we investigate the performance when assigning distinct LLMs to the Actor and State Updater. Specifically, we conduct a cross-evaluation where each module is instantiated by one of three representative models: Qwen-2.5-7B, Qwen-2.5-32B, and GPT-4o. The results in Figure 7 reveal two insights: (1) The State Updater serves as the primary reasoning bottleneck. High-fidelity state tracking requires a capable model (e.g., GPT-4o); utilizing a weaker model here significantly degrades performance regardless of the Actor's strength. This indicates that accurate state abstraction is a fundamental prerequisite for effective downstream reasoning. (2) Hybrid strategies optimize the cost-performance ratio. For instance, pairing a GPT-4o Updater with a lightweight Actor (e.g., Qwen-2.5-7B) achieves success rates comparable to the homogenous GPT-4o setup but at a fraction of the token cost. This suggests that allocating expensive compute resources solely to state synthesis simplifies the decision space for smaller models to execute effectively, enabling strategic efficiency scaling. A detailed efficiency breakdown is provided in Appendix C.

**Case Study.** To deepen our understanding of **ZipAct**'s internal dynamics, we directly examine the execution trace for the ALFWorld task "put a clean soapbar in countertop" shown in Figure 8. From the trace, we observe that **ZipAct** dynamically updates its structured state to capture critical environmental changes. First, upon opening Cabinet 1 and encountering a distractor ("soapbottle 2"), the World State ($\mathcal{W}$) is patched to record this *Entity Map*, while the Goal State ($\mathcal{G}$) maintains persistent focus on the "soapbar," preventing

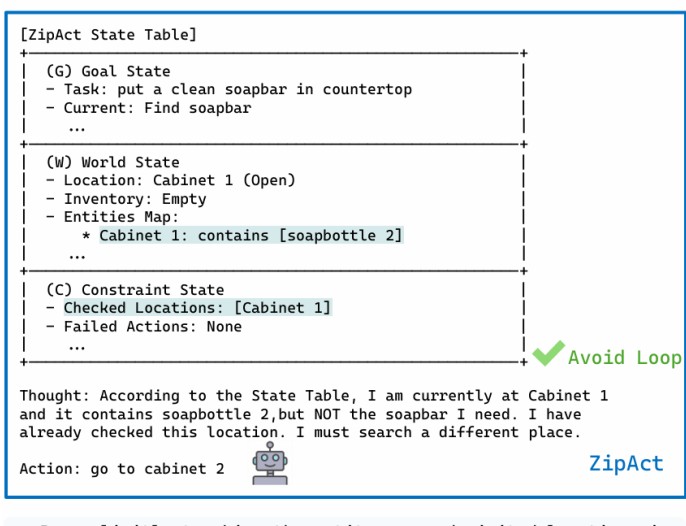

Figure 8: Case study on the ALFWorld 'soapbar' task. By tracking Checked Locations in the Constraint State, the agent identifies Cabinet 1 as visited, effectively pruning the invalid search space.

the agent from being sidetracked by irrelevant objects. Next, recognizing the target is missing, the system updates the Constraint State ($\mathcal{C}$) to explicitly mark "Cabinet 1" as a *Checked Location*. Finally, the Actor selects the most appropriate action by referencing the table—"I have already checked this location"—and correctly decides to move to Cabinet 2. All in all, this demonstrates how `ZipAct` shifts the reasoning paradigm from passive history reading to active state management. By maintaining context at the goal, world, and constraint levels, the framework understands the changing environment and proactively prunes invalid search spaces. In contrast, previous history-dependent methods often suffer from the context snowball effect, where diluted attention leads to redundant hallucination loops.

## 5 Related Work

### 5.1 State-based Agent Reasoning

Drawing inspiration from Chain-of-Thought reasoning (Wei et al., 2022), ReAct (Yao et al., 2023b) introduces a framework that interleaves reasoning traces with actions. Subsequent enhancements seek to improve this backbone through various mechanisms: Reflexion (Shinn et al., 2023) incorporates iterative self-reflection on failures; HiPlan (Li et al., 2025), HyperTree (Gui et al., 2025), and HIMA (Ahn et al., 2025) leverage hierarchical planning and milestone-level guidance; and RAFA (Liu et al., 2024c) evaluates possible future trajectories. Additionally, search-based architectures such as Tree of Thoughts (Yao et al., 2023a) and LATS (Zhou et al., 2024a) extend linear reasoning into tree structures to explore diverse problem-solving paths. Specifically, ReflAct (Kim et al., 2025) and StateAct (Rozanov & Rei, 2025) focus on explicit state tracking to ensure goal alignment and world-grounding. However, these approaches append state or reflection summaries to an ever-growing trajectory. This reliance on cumulative history leads to the context snowball effect. In contrast to these limitations, `ZipAct` fundamentally shifts the paradigm to a state-dependent approach, demonstrating that a compact, structured state is sufficient for achieving robust long-horizon reasoning.

## 5.2 Context Management for LLM Agents

Various strategies have been explored to manage the computational overhead of long-context agents. Heuristic mechanisms, such as token masking (Lindenbauer et al., 2025), trajectory pruning (Zhang et al., 2023; Xiao et al., 2025), and observation truncation in SWE-agent (Yang et al., 2024b), reduce costs but often risk losing critical information. More sophisticated solutions include context distillation like ACON (Kang et al., 2025), belief-bottleneck agents such as ABBEL (Lidayan et al., 2025), memory-augmented systems such as MemGPT (Packer et al., 2023), LightMem (Fang et al., 2026), and GAM (Yan et al., 2025), or test-time optimization via plan caching (Zhang et al., 2025) and action compression (Yuan et al., 2025).Recent attention-based optimizations like SnapKV (Li et al., 2024) and PyramidKV (Cai et al., 2024) attempt to compress the Key-Value cache dynamically to sustain long-context performance. While these methods are ad-hoc or require expensive model-specific training (white-box dependence), **ZipAct** provides a semantic, training-free state update mechanism. It effectively mitigates the challenges of context growth, ultimately serving as a lightweight efficiency booster for various LLM agents. Expanded related works are in Appendix A.

## 6 Conclusion

We proposed **ZipAct**, a state-dependent framework that effectively addresses the context snowball bottleneck in LLM agents. While conventional paradigms like ReAct rely on accumulating raw history, they inevitably suffer from quadratic computational costs ($O(T^2)$) and cognitive distraction. In contrast, **ZipAct** decouples the reasoning process into a memoryless Actor and a semantic State Updater. By zipping verbose history into a compact, unified state comprising Goal, World, and Constraint components, our framework achieves linear complexity ($O(T)$) under a bounded-state assumption and maintains focused reasoning throughout long trajectories. Our extensive experiments across ALFWorld, ScienceWorld, and WebShop demonstrate that **ZipAct** drastically reduces token consumption by over 60% while preserving or improving success rates. By explicitly pruning invalid search spaces and tracking environment dynamics, it establishes a favorable balance between efficiency and performance across different models. We hope this work inspires further research into a shift from passive history accumulation to active state management for complex long-horizon tasks.

## 7 Limitations

While **ZipAct** significantly reduces computational overhead, it relies on a predetermined state structure. This design is most natural for tasks with partially structured state, and may be less flexible than unstructured raw text when capturing highly implicit nuances in open-ended tasks (Xu et al., 2024). More generally, the current G-W-C schema may not transfer cleanly to every domain, and broader generalization may require adaptive schema design or learned state abstractions. Additionally, although **ZipAct** effectively filters noise, there is a natural balance between maintaining a compact state and retaining the exact raw history. Exploring dynamic mechanisms to adjust this balance for specific scenarios is a promising direction for future research (Luo et al., 2025). Finally, our current evaluation is limited to text-based environments. Extending **ZipAct** to multimodal agents is an important next step: rather than simply storing raw perceptual observations, a multimodal version would likely require structured abstractions over visual inputs that can be integrated into the World State ($\mathcal{W}$) in a compact and reliable way.

**Broader Impact.** By reducing token usage in long-horizon LLM agents, **ZipAct** may lower inference cost and energy consumption, which can improve accessibility and deployment efficiency. At the same time, cheaper agent execution may also lower the cost of scaling automated behavior if such systems are deployed without appropriate oversight. Our evaluation is limited to sandboxed text environments, and we do not study real-world deployment risks in this work.

### Acknowledgments

Ming Zhang is supported by the National Key Research and Development Program of China with Grant No. 2023YFC3341203 as well as the National Natural Science Foundation of China with Grant Number

62306014. The authors are grateful to the anonymous reviewers for critically reading this article and for giving important suggestions to improve this article.

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

# A   Expanded Related Works

## A.1   Efficient Context Management

Handling the dichotomy between infinite interaction histories and finite context windows remains a central challenge in LLM agents. To address this, heuristic approaches like Token Pruning and Masking (Lindenbauer

et al., 2025; Zhang et al., 2023; Xiao et al., 2025) reduce computational costs by selectively discarding older or less informative tokens. Recent advancements have pushed this further: StreamingLLM (Xiao et al., 2024b) utilizes attention sinks for stable infinite generation, while PyramidKV (Cai et al., 2024) and SnapKV (Li et al., 2024) dynamically compress key-value caches by identifying crucial attention features across layers. Addressing the memory bottleneck directly, KIVI (Liu et al., 2024d) introduces a tuning-free 2-bit quantization specifically for KV caches, and Quest (Tang et al., 2024) proposes a query-aware sparsity mechanism to accelerate long-context inference. A more structural alternative is Memory Augmentation, exemplified by systems like MemGPT (Packer et al., 2023), LightMem (Fang et al., 2026), and InfLLM (Xiao et al., 2024a), which manage a "virtual context" by paging information between a limited context window (RAM) and external vector storage. While theoretically supporting infinite contexts, these systems often introduce retrieval latency. Parallel research explores Context Compression: techniques like LLMLingua (Jiang et al., 2023) discard low-perplexity tokens, whereas Context Distillation methods like ACON (Kang et al., 2025) generate abstract summaries. Unlike these approaches, which often operate at a superficial token level or risk losing task-critical entities, **ZipAct** employs a semantic, state-dependent paradigm to maintain a compact structured representation whose per-step size remains stable in our evaluated settings.

## A.2 Advanced Planning in Agents

Effective planning is a prerequisite for long-horizon tasks, necessitating frameworks beyond linear chain-of-thought (Yao et al., 2023b). Reflective Agents such as Reflexion (Shinn et al., 2023) and ReflAct (Kim et al., 2025) use verbal reinforcement loops to correct failures. However, accumulating verbose reflections often exacerbates the context snowball effect. To enhance reasoning structure, Tree-Search Planning methods like Tree of Thoughts (ToT) (Yao et al., 2023a) and Graph of Thoughts (GoT) (Besta et al., 2024) have been adapted for agents. Notable examples include LATS (Zhou et al., 2024a), which unifies reasoning with Monte Carlo Tree Search, and Algorithm of Thoughts (AoT) (Sel et al., 2024), which propels LLMs through algorithmic pathways in-context. Similarly, RAP (Hao et al., 2023) leverages a learned world model to simulate future states. While powerful, these search-intensive methods are often computationally prohibitive. Alternatively, Self-Discover (Zhou et al., 2024b) allows LLMs to self-compose reasoning structures for complex tasks. In the realm of Hierarchical Planning, methods like HiPlan (Li et al., 2025) and AdaPlanner (Sun et al., 2023) decompose goals to manage long horizons. Finally, inspired by dual-process theory, SwiftSage (Lin et al., 2023) combines fast intuitive thinking with slow deliberate planning. **ZipAct** synthesizes these insights by explicitly incorporating a *Goal State* ($\mathcal{G}$) to track sub-goal progression, but unlike prior works that rely on complex multi-agent setups or ever-growing contexts, **ZipAct** integrates this tracking directly into a unified, linear-cost state update loop.

## B Evaluation Details

All methods are evaluated with a maximum horizon of 50 interaction steps per episode, and unfinished episodes are counted as failures. Across the evaluated episodes, the average realized interaction lengths are 12.5 for ALFWorld, 14.2 for ScienceWorld, and 11.3 for WebShop. Unless otherwise noted, decoding uses deterministic generation with temperature = 0.0. For ScienceWorld, binary success rate is used as the primary evaluation metric in the main results.

## C Efficiency Breakdown

Our cumulative token counts include both input/prompt and output/completion tokens across all LLM calls. ReAct uses one action-generation call per interaction step. ZipAct uses one initialization call, then one Actor call per step, plus one Updater call after each executed action from the second step onward. Relative to ReAct, ZipAct becomes cheaper after 6.1 interaction steps on average over the evaluated episodes.

For proprietary backbones, we additionally report rough per-episode API cost estimates. For GPT-4o-mini, the average per-episode cost across ReAct / Mask / Summary / Reflexion / ZipAct is approximately 0.031 / 0.017 / 0.016 / 0.043 / 0.012 US dollars. For GPT-4o, the corresponding values are approximately 0.530 / 0.296 / 0.289 / 0.796 / 0.172 US dollars. For the open-source Qwen backbones, deployment-dependent

Table 4: Comparison between StateAct and **ZipAct** under the same evaluation settings.

| Backbone | Method | Avg. SR | Avg. Tokens |
|---|---|---|---|
| GPT-4o-mini | StateAct | 40.2 | 182k |
| | **ZipAct** | **44.0** | **62k** |
| GPT-4o | StateAct | 58.8 | 186k |
| | **ZipAct** | **61.3** | **53k** |
| Qwen-2.5-7B | StateAct | 35.8 | 167k |
| | **ZipAct** | **38.9** | **58k** |
| Qwen-2.5-32B | StateAct | 56.1 | 181k |
| | **ZipAct** | **58.1** | **55k** |

serving cost varies substantially, so we report token usage rather than provider-specific API cost. Lower token usage does not necessarily imply lower end-to-end latency, since latency also depends on the number of model calls and provider-side overhead.

## D   Baseline Implementation Details

For the masking baseline, we keep the full action history but replace observations older than the most recent five with the literal placeholder `[Observation masked]` (`keep_recent = 5`). Action generation reuses the same environment-specific ReAct prompt/template as the ReAct baseline, with temperature $= 0.0$ and `max_tokens = 256`.

For the summarization baseline, we use `summary_interval = 10`. The summary prompt requests a concise 3–5 sentence summary of explored areas, found objects, attempted actions, and current progress. Summary generation uses temperature $= 0.0$ and `max_tokens = 300`. Action generation again reuses the same environment-specific ReAct prompt with temperature $= 0.0$ and `max_tokens = 256`, conditioning on the running summary plus the most recent `2 * summary_interval` history items.

## E   Additional Comparison with StateAct

Across all four overlapping backbones, **ZipAct** remains better than StateAct in both average success rate and token efficiency. Notably, StateAct uses more cumulative tokens than the ReAct baseline reported in the main paper, suggesting that appending explicit state descriptions to an ever-growing trajectory does not by itself alleviate the context snowball issue. We do not include ReflAct as a direct baseline because its code is not publicly available, and we discuss ACON, LightMem, and MemGPT as relevant related approaches rather than direct training-free baselines for our setting.

## F   Prompt

The prompt templates and injectables table are in §3.4 of the main text. The full prompt templates used in our experiments, including the environment-specific ReAct and ZipAct prompts, are provided in the supplementary material. The full JSON output schema for the State Updater is reproduced below for reference.

```
{
  "goal_state": {
    "global_instruction": "<immutable_main_task>",
    "sub_goal_queue": ["<next_sub_goal>", ...],
    "current_objective": "<active_sub_goal>"
  },
  "world_state": {
    "location": "<current_location>",
    "inventory": ["<item_1>", ...],
```

```
    "entity_map": { "<object_id>": "<state_description>" }
  },
  "constraint_state": {
    "negative_constraints": ["<failure_reason>"],
    "visited_locations": ["<loc_1>", ...]
  }
}
```

## G  Updater Failure Analysis

We categorize representative State Updater errors into three types. *State corruption* refers to updates that introduce incorrect or internally inconsistent world-state information, such as wrong location, inventory, or entity status. *Goal drift* refers to cases where the updater mis-tracks task progress, for example by advancing the sub-goal queue prematurely, dropping the current objective, or retaining an outdated objective after the environment has changed. *Action-grounding errors* refer to cases where the updated state no longer supports valid next-step action selection, typically because relevant affordances, constraints, or previously established facts are missing, misinterpreted, or linked to the wrong entities. In our analyzed updater errors, these categories account for 52%, 26%, and 22%, respectively. When the updater output cannot be parsed as valid JSON, we fall back to the previous state; the three categories above therefore focus on semantically incorrect updates that pass parsing. We do not currently use an additional verifier, retry mechanism, or semantic correction stage, and we do not yet report a separate controlled sensitivity-to-error study.

