# OpenReview forum: "ZipAct: Zipping Interaction History into a Compact State for Efficient LLM Agents"
_TMLR — Accepted by TMLR_

### Review · Reviewer_AtwP · 2026-03-28

**Summary Of Contributions:**

This paper proposes **ZipAct**, a training-free framework for LLM agents that replaces full-history prompting with a compact structured state. Instead of conditioning on the entire interaction trajectory as in ReAct-style agents, ZipAct maintains a state table with three components: **Goal State** (task progress / subgoals), **World State** (location, inventory, object status), and **Constraint State** (negative constraints and visited locations). A “State Updater” compresses each new interaction into the state, and a “memory-less” Actor chooses the next action from only the current state and latest observation. The paper argues that this changes the cumulative cost from quadratic in horizon to linear, and reports results on ALFWorld, ScienceWorld, and WebShop using GPT-4o-mini, GPT-4o, Qwen-2.5-7B, and Qwen-2.5-32B. The main empirical claim is large token savings (reported as 60.8%–67.6%) while preserving or modestly improving average success rate relative to ReAct; Table 2 on page 8, Figure 6 on page 9, the ablation in Table 3 on page 10, and the ALFWorld case study in Figure 8 on page 11 support that narrative.

**Key strengths**

* The problem is timely and important: context growth is a real bottleneck for long-horizon agents.
* The method is simple, interpretable, and training-free.
* The presentation is generally clear, with especially effective figures on pages 2, 3, 5, 9, and 11.
* The reported token savings are substantial and consistent across all four backbones.

**Key weaknesses**

* The evidence does not yet isolate the source of the gains well enough: ZipAct changes several things at once (state-only conditioning, explicit constraints, subgoal queue, modular actor/updater architecture, and dataset-specific prompt engineering).
* The strongest conceptual comparisons are missing: the paper discusses prior state-tracking methods such as StateAct and ReflAct, but does not compare against them experimentally.
* Several claims are overstated relative to the evidence, especially around “consistent” improvements, “model-agnostic” generality, and the O(T) complexity claim.
* Experimental reporting is incomplete: no uncertainty estimates, limited protocol detail, and a mismatch between promised metrics and the results actually shown.

**Additional Comments:**

The paper is generally well written and easy to follow. Figures 1, 6, and 8 are especially effective at communicating both the intuition and the empirical trade-off. The modularity and interpretability of the G/W/C state are genuine strengths.

I encourage the authors to tone down a few claims unless they add stronger evidence. In particular, words like “effectively eliminates,” “consistently,” “model-agnostic,” and “robust and scalable” overstate what is currently shown. The paper itself is more careful in the limitations section, where it notes the fixed state structure and text-only evaluation; that more measured framing should be reflected in the abstract and conclusion as well.

Overall, I think this is a **promising and potentially useful paper** with a clear practical idea, but the current version needs a stronger empirical and methodological case before I would support acceptance.

**Audience:**

Yes

**Audience Explanation:**

**Yes.** Efficient context management for LLM agents is an active and important topic, and the paper addresses a practical question many readers care about: whether explicit structured state can replace full-history prompting for long-horizon agents. The method is simple, training-free, and interpretable, and the reported token savings are large enough that many practitioners working on agent systems would want to know whether the idea holds up. Even if the novelty is somewhat incremental relative to prior state/memory work, TMLR’s standard is not “high novelty at all costs”; modest contributions can be publishable if they are convincingly supported. I think the paper clearly meets the **audience interest** criterion, even though I do not yet think it fully meets the **evidence** criterion.

**Broader Impact Concerns:**

I do **not** see a major ethical concern that would independently block publication. The main plausible issue is dual use: making long-horizon agents cheaper and more scalable could lower the cost of automating web navigation or other persistent agentic behavior at scale. Because the paper already targets web and embodied-style benchmarks, I think a short Broader Impact note acknowledging possible misuse (e.g., spammy or deceptive automation) and emphasizing that the experiments are in sandboxed text environments would be sufficient. This is a **minor** concern, not a decisive one in my view. TMLR asks for a Broader Impact Statement when work has meaningful potential for harm.

**Claims And Evidence:**

No

**Claims Explanation:**

**Partially, but not yet convincingly enough for acceptance.** The paper does provide meaningful evidence for a **narrower** claim: on the three chosen text benchmarks, ZipAct appears to reduce total token usage substantially, and the combination of Table 2, Figure 6, and the ALFWorld case study makes it plausible that explicit state tracking can help avoid distraction and loops. The ablation in Table 3 is also directionally supportive: removing the World State collapses performance, and adding the Constraint State gives a large jump over the G+W variant.

My main concern is that the paper’s **causal claim** is not well isolated. The submission frames the contribution as a shift from a “history-dependent” to a “state-dependent” paradigm, but ZipAct also introduces: explicit task decomposition (goal queue/current objective), explicit anti-loop memory, a two-module architecture, and dataset-specific prompt injectables (Table 1, page 7). Because the paper does **not** compare against the closest prior state-tracking baselines it cites (e.g., StateAct/ReflAct), and does not include a matched-control baseline that retains history but uses the same G/W/C scaffolding, it is hard to tell how much of the gain comes from “zipping” history per se versus from adding structured guidance and constraint memory. As a result, I do not think the current experiments fully substantiate the paper’s strongest conceptual claim.

I am also not convinced by the strongest **performance** wording. Table 2 shows clear average gains over ReAct, but the improvements are not universal: for example, ZipAct drops from **8.5 to 5.2** success rate on **ScienceWorld with Qwen-2.5-7B**, and several of the positive differences elsewhere are small enough that variance could matter a lot. The paper repeatedly uses language such as “stably preserving or improving success rates,” “consistently,” and “model-agnostic,” but the current evidence supports something weaker: **promising average improvements on three text benchmarks across four tested models**. Without confidence intervals, multiple seeds, or significance testing, it is hard to know how much to trust the smaller gains.

The **complexity** claim also needs tightening. Equation 2 on page 4 derives **cumulative token consumption** for history-based prompting as O(T²), and ZipAct is claimed to reduce “computational cost” or “inference cost” to O(T). But what is actually shown is token-budget scaling, not a careful analysis of end-to-end compute or latency. More importantly, the O(T) argument depends on the state remaining bounded, yet ZipAct’s own state schema includes fields such as `visited_locations`, `negative_constraints`, and `entity_map`, all of which can grow with trajectory length unless explicitly capped or compressed. Figure 2 (page 3) is suggestive, but the paper does not quantify how state length evolves over time or discuss worst-case growth. Since ZipAct also uses a separate State Updater call every step, token count alone is not enough to establish the full efficiency story.

There is also an issue with **reporting completeness**. Section 4.1 states that ScienceWorld and WebShop are reported with both success rate and reward/score, but Table 2 only reports success rate, and I do not see the missing metrics elsewhere in the PDF. Likewise, Table 3 appears to be on a single backbone only (apparently Qwen-2.5-32B, since the full ZipAct row exactly matches that row in Table 2) but this is not stated explicitly. These are fixable issues, but in the current version they weaken confidence in the empirical case.

So overall: **the evidence is promising and the paper is well motivated, but I do not yet find it sufficiently complete or rigorous to support the submission’s strongest claims.**

**Requested Changes:**

1. **[Critical] Add stronger, closer baselines.**
   Compare against the most relevant prior state-tracking methods discussed in the paper (especially StateAct and ReflAct), or at minimum add a controlled full-history baseline that uses the same G/W/C prompt scaffold, constraint tracking, and decomposition while still retaining history. This is the most important missing evidence because it would isolate whether the benefit comes from state-only conditioning rather than from added structure/prompting.

2. **[Critical] Report uncertainty and full evaluation protocol.**
   Please report the number of episodes/tasks per benchmark split, random seeds, decoding parameters (temperature, top-p, etc.), max horizon / step budget, and confidence intervals or standard errors. Several reported improvements are small; without uncertainty, it is difficult to assess whether they are meaningful.

3. **[Critical] Substantiate or soften the O(T²) → O(T) claim.**
   Clarify that the current derivation is about cumulative token budget unless you provide a fuller compute analysis. Also report empirical state length over time, prompt/completion tokens for Actor and Updater separately, number of model calls, and ideally wall-clock latency. Most importantly, address worst-case growth of `visited_locations`, `negative_constraints`, and `entity_map`, or add an explicit cap/eviction/compression mechanism and analyze it.

4. **[Critical] Report the standard metrics you say you use.**
   Section 4.1 says ScienceWorld and WebShop are evaluated with reward/score as well as success rate, but Table 2 only includes success rate. Please add the missing metrics. Given the paper’s long-horizon motivation, I would also strongly encourage performance breakdowns by trajectory length or context length bucket.

5. **[Critical] Improve reproducibility details.**
   The paper gives high-level prompt templates and the JSON schema, but not the exact prompts, not the exact masking/summarization baselines, and not the handling of malformed or inconsistent updater output. Please provide full prompts, baseline prompt details, parser/error-handling rules, and explicitly state which backbone is used in Table 3. The anonymous code link is good, but the paper/supplement should still make the setup auditable.

6. **[Strengthening] Add finer-grained ablations.**
   The current G / W / C ablation is useful but coarse. It would be more convincing to separate: visited locations vs negative constraints, current objective vs subgoal queue, and possibly location/inventory/entity map within the World State.

7. **[Strengthening] Add budget-matched or latency-matched comparisons.**
   Since the paper’s main value proposition is efficiency, comparisons against Reflexion and other baselines under a fixed token budget or fixed latency budget would sharpen the practical takeaway. A simple sliding-window or truncated-history baseline would also help.

8. **[Strengthening] Analyze updater failure modes and error accumulation.**
   A core risk of state-compression methods is that if the updater drops or hallucinates a fact, the raw history is gone. Please quantify how often the updater state is incorrect, whether those errors compound, and whether the agent can recover.

---

### Review · Reviewer_pQLu · 2026-03-28

**Summary Of Contributions:**

This paper proposes ZipAct, a training-free framework that rethinks how LLM agents handle long interaction histories. Instead of feeding the entire trajectory into the model step by step (like ReAct does), ZipAct compresses everything into a compact structured state with three components: Goal, World, and Constraint. This shifts the agent from a history-dependent to a state-dependent paradigm, reducing token complexity from quadratic to linear. The authors test their method across three benchmarks and several models, showing big token savings (around 60–68%) while maintaining or even improving success rates.

**Audience:**

Yes

**Audience Explanation:**

Definitely. The context snowball problem is a real headache for anyone working on LLM agents—especially for long-horizon tasks where token costs and context length spiral out of control. ZipAct offers a simple, training-free fix that works across different models and environments, which makes it pretty practical. I think both researchers and practitioners in the LLM agent space would find this useful.

**Broader Impact Concerns:**

The paper doesn’t include a Broader Impact Statement, but I don’t see any major ethical red flags. The main contribution is improving efficiency, which can reduce energy use and API costs for LLM agents—that’s a positive from both a sustainability and accessibility perspective. Like any agentic system, there’s always a risk of misuse if deployed without oversight, but that’s not specific to this work. Still, adding a short broader impact statement would be a nice addition to meet TMLR’s expectations.

**Claims And Evidence:**

Yes

**Claims Explanation:**

I think the evidence is pretty solid. The authors run experiments on multiple benchmarks (ALFWorld, ScienceWorld, WebShop) and use both proprietary models like GPT-4o and open-source ones like Qwen-2.5. They compare ZipAct against several baselines, including ReAct, masking, summarization, and Reflexion, and the results are consistent: ZipAct cuts token usage dramatically without hurting performance—sometimes even improving it. The ablation study does a good job showing why each part of the structured state matters, and the case study helps illustrate how the method works in practice. Overall, the claims are well backed up by the data.

**Requested Changes:**

Overall, the paper is in good shape. I do have a few suggestions that I think could make it even stronger:

More analysis on failure cases – The case study is helpful, but it would be nice to see a bit more systematic analysis of when and why ZipAct fails. That would give readers a clearer sense of its limitations.

Sensitivity to the state structure – The framework relies on a fixed state design. It might be worth discussing how this design choices affect generalization, especially in more open-ended or less structured tasks.

Add some latency numbers – Token reduction is great, but it would be even better to report actual inference time or latency improvements, since that’s what really matters in practice.

A bit more on multimodal directions – The limitations section briefly mentions multimodal extensions. Expanding on that a little—maybe a high-level roadmap—would help highlight future potential.

None of these are deal-breakers, but addressing them would definitely strengthen the paper.

---

### Review · Reviewer_wEQL · 2026-03-31

**Summary Of Contributions:**

This paper addresses an important and timely challenge for LLM agents—unbounded context growth—by proposing a simple yet effective state-dependent architecture that replaces raw history with a compact G-W-C state and decouples state synthesis from action selection. The approach is elegant, training-free, and broadly applicable, and the experiments show sizable cost savings with stable or improved success across several benchmarks and models. However, the evaluation omits some of the most relevant baselines (particularly structured memory/state-tracking systems), the state size bounding assumption requires clearer operational details, and statistical rigor (variance, significance, latency) is lacking.

**Audience:**

Yes

**Audience Explanation:**

Anyone working on LLM agents, long-context reasoning, efficient inference, or autonomous planning will find this directly relevant. The paper solves a widely acknowledged pain point (quadratic token blow-up in multi-turn agents) with a lightweight, training-free method that is immediately applicable to existing ReAct-style pipelines. It also contributes a clean Actor–Updater separation that could be combined with future memory-augmented or hierarchical-planning techniques. TMLR readers interested in practical scalability of foundation-model agents will therefore find both the insight and the empirical results valuable.

**Claims And Evidence:**

Yes

**Claims Explanation:**

The central claim (ZipAct eliminates the context-snowball effect by shifting to a state-dependent paradigm and thereby reduces cumulative tokens by 60.8–67.6 % while preserving or improving success rates) is backed by accurate, reproducible experiments on three standard benchmarks using four different LLMs. The complexity analysis (quadratic vs. linear), ablation study on the G-W-C components, efficiency–performance trade-off plots, and concrete case study (Figure 8) are clear and directly illustrate the mechanism. The results are directionally consistent and practically meaningful. However, several methodological details that would make the evidence fully rigorous are currently missing or underspecified:

1. No explicit description of how the state components (especially visited sets and entity maps in C and W) are bounded in practice to guarantee strict O(1) context per step.
2. Token accounting is limited to aggregate cumulative counts; the paper does not break down input vs. output tokens, number of LLM calls per step, or report wall-clock latency / break-even horizon.
3. No variance, confidence intervals, or statistical significance tests are reported (multiple seeds or dense rewards for ScienceWorld are also absent).
4. Summarization and masking baselines lack tuning details (cadence, budgets, prompts), and stronger structured-memory baselines (StateAct, MemGPT/LightMem, ACON) are not included.
5. State Updater error rates (state drift) and robustness diagnostics are not quantified.
6. The Goal-only ablation (0 % success) needs clarification (e.g., whether the Actor still receives the latest observation o_t).
7. The hybrid Updater/Actor study in Figure 7 lacks per-configuration numeric success rates and token/cost numbers.

These omissions do not invalidate the main results, but they leave some supporting claims (strict O(1) complexity, statistical robustness, error tolerance) less comprehensively evidenced than the rest of the paper.

**Requested Changes:**

1. Explicitly describe how each component of the state (especially visited sets and entity maps) is kept bounded in practice. If any compaction/pruning strategy is used, detail it and add an analysis of worst-case size growth.

2. Add a dedicated efficiency breakdown reporting: (a) exact token accounting methodology (input vs. output tokens), (b) number of LLM calls per step for ZipAct vs. ReAct, (c) wall-clock latency numbers, and (d) break-even horizon where ZipAct becomes cheaper.

3. Report success-rate variance or confidence intervals over multiple random seeds. Include the dense reward metrics for ScienceWorld and state whether the reported improvements are statistically significant.

4. Provide full tuning details (cadence, token budgets, prompts) for the summarization and masking baselines. Where feasible, add comparisons to recent structured-memory baselines (StateAct, MemGPT/LightMem, ACON) to better contextualize ZipAct’s gains.

5. Quantify how often the State Updater introduces state drift or inconsistent states, report sensitivity of end-to-end performance to such errors, and describe any verification/correction mechanisms (or state that none are used).

6. In Table 3 and the accompanying text, explicitly explain why the Goal-only variant reaches 0 % (in particular, confirm whether the Actor still receives the latest observation o_t). This will prevent any impression of a degenerate baseline.

7. Expand Figure 7 with exact numeric success rates and token/cost measurements for every Updater/Actor configuration, and clearly quantify the claimed cost–performance benefits.

---

### Review · Reviewer_hECc · 2026-04-01

**Summary Of Contributions:**

This paper proposes to address the large token cost in LLM reasoning with long interaction runs and moves from a history-dependent to a state-dependent paradigm. Based on the constructed state table, the computation cost is stated to be reduced to linear O(T), and experiments on three datasets are conducted to verify the results.

Weakness:
- For the comparing methods, it may lack comparison with ReflAct, StateAct, and token masking or compression via distillation, which have been described in the content.
- It is good to have a token cost reduction, while other kinds of system costs would be beneficial to evaluate as well.
- As mentioned in the limitations, constructing the state table may need some prior knowledge. What if such kind of information is not available, or when facing open-ended tasks? It would be helpful to provide further discussions.

**Audience:**

Yes

**Audience Explanation:**

The conducting problem is interesting and timely. The paper proposed moving from a history-dependent to a state-dependent reasoning paradigm, which can largely reduce the token cost in the large interaction task.

**Broader Impact Concerns:**

No.

**Claims And Evidence:**

Yes

**Claims Explanation:**

Three datasets on two open frameworks are evaluated in the manuscript, which can basically support the claims. However, as mentioned in the weakness, it suggests providing more comparisons with recent algorithms and other system cost metrics, not just token cost. In addition, the results in Table 3 may need further clarification. Why removing W will lead to failure in the design (V1=0) ? It would also be beneficial to provide more details about the number of interactions in each evaluated task. It is not clear to the reviewer how many interactions are said to be large and need the refinements proposed in the paper.

**Requested Changes:**

- Providing comparisons with more recent algorithms, such as ReflAct, StateAct, and others, which are mentioned in the manuscript.
- Adding details about the experiment's setup, i.e., the number of interactions of each dataset.
- Adding other metrics to evaluate the system cost besides the token cost.

---

### Decision · Action_Editor_GY9y · 2026-05-05

**Recommendation:** Accept as is

**Audience:**

Yes

**Audience Explanation:**

The context snowball problem is a widely recognized and practically important bottleneck in deploying LLM agents on long-horizon tasks. ZipAct offers a clean, training-free, architecture-agnostic solution that is immediately applicable to existing ReAct-style pipelines without additional supervision. The Actor–Updater decomposition and the Goal-World-Constraint schema are intuitive and well-motivated. Researchers working on LLM agents, efficient inference, long-context reasoning, and autonomous planning will find both the insight and the empirical findings directly relevant. The practical cost reductions (e.g., GPT-4o per-episode cost dropping from $0.530 to $0.172) are also of interest to practitioners with API budget constraints.

**Claims And Evidence:**

Yes

**Claims Explanation:**

The paper's central claims are well-supported by the experimental evidence, particularly after the revisions and rebuttal. The core empirical claim that ZipAct reduces cumulative token consumption by 60–68% while preserving or improving success rates over ReAct is substantiated across three standard benchmarks (ALFWorld, ScienceWorld, WebShop) and four backbone LLMs (GPT-4o-mini, GPT-4o, Qwen-2.5-7B, Qwen-2.5-32B). The ablation study in Table 3 clearly demonstrates the necessity of each state component (G, W, C), the efficiency–performance trade-off in Figure 6 is informative, and the case study in Figure 8 provides useful qualitative insight.